# Comparison of Nonlinear Growth Models to Estimate Growth Curves in Kivircik Sheep under a Semi-Intensive Production System

**DOI:** 10.3390/ani13142379

**Published:** 2023-07-21

**Authors:** Nursen Ozturk, Pembe Dilara Kecici, Lorenzo Serva, Bulent Ekiz, Luisa Magrin

**Affiliations:** 1Department of Animal Breeding and Husbandry, Faculty of Veterinary Medicine, Istanbul University-Cerrahpasa, 34500 Istanbul, Turkey; dilara.kecici@iuc.edu.tr (P.D.K.); bekiz@iuc.edu.tr (B.E.); 2Department of Animal Medicine, Production and Health, University of Padova, 35122 Padova, Italy; lorenzo.serva@unipd.it (L.S.); luisa.magrin@unipd.it (L.M.)

**Keywords:** Kivircik lamb, growth curves, nonlinear models, Gompertz model, maturity rate

## Abstract

**Simple Summary:**

Following the growth of a fattening animal is an important aspect since farmers aim to maximize their profits by improving their slaughter weight. Growth models can fit the age–weight data into biologically interpretable parameters that provide a growth curve for monitoring animal growth, thus identifying the growth declining point during the fattening period. In this study, we aimed to define the best fitting growth model for describing Kivircik lamb growth. While the Gompertz model was identified as the best fitting growth model, it was determined that male lambs, twin lambs, and winter-born lambs had a higher mature weight. Even though they may be lighter at birth, the proportion of their prospective weight gain was found to be higher compared to female, single-born, and spring-born lambs. Our results showed that the estimation error of the Gompertz model ranged between −0.43 kg to 0.60 kg. In conclusion, according to the Gompertz model’s estimation, Kivircik lambs reached 40.7% of their slaughter weight at weaning and had a 66.3% degree of maturity at the slaughter age, which indicated inappropriate management.

**Abstract:**

The Kivircik is an indigenous sheep breed from Turkey, and it has superior meat quality compared to other indigenous breeds. Therefore, farmers prioritize Kivircik lamb fattening instead of milk production. Here, we aimed to determine the best nonlinear growth model, i.e., Gompertz, Logistic, Von Bertalanffy, and Brody, to describe the growth curve of Kivircik lambs. The body weight data from birth until 150 days of age belonging to 612 lambs were used as the material of this study. The best fitting model was selected by considering the adjusted coefficient of determination (R^2^_adj_), residual mean square, and Akaike’s (AIC) and Bayesian information criteria (BIC). Even though the Brody model had a better statistical fit, considering its biological interpretation, the Gompertz model was identified as an appropriate model for describing Kivircik lamb growth. Male lambs, twin lambs, and lambs born in winter had higher mature live weights (44.2 kg, 71.2 kg, and 38.5 kg, respectively) and rate of weight gain (2.1, 2.6, and 2.0, respectively). However, our subgroups revealed a similar rate of maturity (0.01). Growth models are important tools for deciding the optimal slaughter age and they provide valuable information on the management practices of both sexes, birth types, and birth seasons. These results can be applied to breeding programs for early selection, enabling intervention strategies when needed.

## 1. Introduction

In livestock production, growth is mainly concerned with the growth of muscle, fat, bone, and mammary glands, which are linked to economically important traits. Therefore, following a farm animal’s growth is significant when farmers aim to maximize their profits by optimizing the slaughter weight and slaughter age to meet various grading requirements. For this aim, recording body weight is the simplest and most widely used technique for monitoring the slaughter age of animals [1].

Recently, the precision farming context that incorporates technology to assist farmers in livestock production has been gaining importance. Since studies revealed that a delay in the optimal slaughter weight is negatively associated with the profitability of the system [2,3,4,5], the ability to estimate the slaughter weight of an animal for a given age would allow farmers to adopt any strategy that would improve their profitability. At this point, growth curves are becoming an important tool for monitoring the age–weight relationship. In particular, nonlinear models help to condense large volumes of longitudinal data into biologically interpretable parameters that would help users understand the potential mature weight, growth rate, and the proportion of weight to be gained (inflection point) [6]. In a review study by Van der Merwe et al. [3], the significance of the mature weight was related to the mature size of an animal, which in turn influenced the growth rate and maturity of the animal. The mature weight has been found to be negatively correlated with the inflection point since the inflection point and growth rate are positively correlated [6]. These parameters are important for managing animal feeding and deciding the optimal slaughter age from an economic point of view since an increase in each kg of mature weight results in increased maintenance cost(s) and a deterioration in feeding efficiency [3] as well as a slow growth rate, resulting in a low slaughter weight that limits profitability [7]. 

Considering the economic importance of small ruminants for the rural economy, studies characterizing lamb growth patterns to increase lamb productivity and boost farmer profitability are of great concern. Simple linear models, models with both exponential and negative exponential growth, and models with a sigmoidal shape (S-shape) are generally applied to fit the age–weight data into growth curves for lambs [8]. However, the best fitting model that appropriately describes lamb growth remains controversial due to different sample sizes, breeds of sheep, management practices, production types, and climates among the studies [9,10,11,12,13,14,15]. Additionally, Van der Merwe et al. [3] identified that the type of data would determine the best fitting growth model. For instance, in datasets with static functions, the Brody model was more successful in predicting lamb growth, while for the growth estimations using continuous data measurements, the Gompertz and/or von Bertalanffy models provided better estimates. In addition, the growth performance varied between the sexes, birth-type groups (singleton or twin), and lambs born in different seasons [13,16,17].

The Kivircik is a native sheep breed localized in the Thrace, Marmara Region and in some regions of Greece and Bulgaria. Kivircik sheep farming is largely based on 93.6% pasture usage where the farmer’s use of the pasture can reach up to 11.4 months/year [18]. This breed has superior meat quality characteristics compared to other indigenous breeds [19], which is one of the reasons for obtaining a geographical indication certification that provides farmers the opportunity to sell their products for a higher price [20]. Due to its importance for the region, two selection projects have been performed to improve the production traits of Kivircik lambs [21]. 

In Turkey, consumer preference for lamb meat is high compared to Mediterranean European countries (i.e., France, Greece, Italy, and Spain) [22]. However, lamb production is mainly conducted using native breeds with moderate growth performance [23]. This implies that the possible transformation of sheep husbandry from extensive to intensive production to meet consumer demands may be imminent. In intensive sheep husbandry, growth models are important for defining the growth patterns of lambs and monitoring their growth in accordance with the aforementioned patterns. When a deviation from the estimated live weight occurs, farmers can identify reasons and solutions with the help of growth curves. Similarly, besides deciding the optimal slaughter age, it would be possible to efficiently manage the operation strategies for various other aspects, including lamb feeding, selection at early stages, and identifying healthy/unhealthy animals [24]. 

In Turkey, growth model studies that were conducted using the Kivircik breed are limited and did not investigate the effects of environmental factors on the growth patterns of Kivircik lambs [25,26]. Considering the importance of Kivircik lamb production for both farmers and consumers, estimating Kivircik lamb growth is required to improve production traits. In this study, we aimed to determine the best nonlinear growth function for predicting Kivircik lamb growth as well as the growth parameters for different sexes, birth types, and seasons of birth. For this aim, we compared four nonlinear growth models (i.e., Gompertz, Logistic, Von Bertalanffy, and Brody) to estimate the best growth curve for Kivircik lambs. These models were selected due to their wider use in the literature, their ability to reveal biologically interpretable parameters, and their relative ease of data fitting for three parameter models [8]. We believe the estimated biological parameters from this study would enhance Kivircik lamb meat production and that they could be considered in the selection criteria for modifying the shape of the Kivircik lamb growth curve. 

## 2. Materials and Methods

### 2.1. Animal Management and Data Collection 

This study was conducted at the Kivircik sheep breeding farm of the University of Istanbul, Faculty of Veterinary Medicine, which was used for experiments and education. The lambs and dams were housed in an individual birth cubicle until the age of seven days. During the day, the dams grazed in the pasture and in the evening they only received alfalfa hay. *Ad libitum* alfalfa hay (87.8% dry matter, 12.9% crude protein, 2.31% crude fat, 37.3% crude fiber, 9.87% ash, 43.9% neutral detergent fiber (NDF), 58.8% acid detergent fiber (ADF), and 1843 MJ ME/kg DM) was supplied after 15 days of age, while concentrate feed containing 89% dry matter, 17% crude protein, 4.82 % crude fat, 6.41% crude fiber, 7.96% ash, 10.2% NDF, 24.6% ADF, 0.82% calcium, 0.51% phosphorus, and 12 MJ ME/kg DM was supplied after one month. The amount of concentrate feed was increased gradually and reached 400 g/lamb per day after weaning. The lambs were weaned at 75 days of age and then they began to graze. The pasture was natural and composed of (on a dry matter basis) 52% *Gramineae* (*Festuca* spp. and *Lolium* spp.), 22% *Leguminosae* (mainly *Trifolium* spp., *Medicago* spp. and *Vicia* spp.), and 26% other families (mainly *Conium* spp., *Geranium* spp., *Viola* spp., *Rumex* spp. and *Plantago* spp.). The chemical composition of the pasture was constituted by 38.0% dry matter, 11.5% crude protein, 5.46% ether extract, 24.4% crude fiber, 10.4% ash, 42.7% NDF, 36.0% ADF, and 9.21 MJ ME/kg DM [27]. 

In order to record the body weights of the Kivircik lambs, the lambs were weighed every fifteen days between 2014–2016. The data belonging to six hundred and twelve lambs from birth to 150 days were used as the material for this study. Due to the preferred slaughter age in the field of 150 days, the body weight data were kept between birth and 150 days of age. The descriptive statistics related to the age–weight records are displayed in Table 1. The mean birth, weaning, and slaughter weights of the Kivircik lambs were determined as 4.36 kg (±0.86 kg), 16.0 kg (±6.43 kg), and 26.0 kg (±5.07 kg), respectively (Table 1).

The number of observations and their frequencies in each subgroup are shown in Table 2. The lambs were distributed evenly in each subgroup, except for the year variable. Since only 17% of the lambs were included in 2015, the year factor was not evaluated in the further analysis that determined the best growth model. 

### 2.2. Statistical Analysis

#### 2.2.1. Exploratory Analysis: Principal Component Analysis

To provide an exploratory analysis, a principal component analysis (PCA) was conducted to visually assess the original explanatory variables and the relationships among the various groups (sex, birth type, and season of birth) in the new features or principal components. These statistics were performed using the XLSTAT software (XLSTAT, Addinsoft, release version 2022.2.1, New York, NY, USA).

#### 2.2.2. Nonlinear Growth Models

To estimate the growth rate age–weight relationship, the collected data from the lambs′ dataset were fitted using the Gompertz (1), Logistic (2), von Bertalanffy (3), and Brody (4) models [28,29,30]. The tested models were defined as the following.
y = A × exp (−B × exp (−K × t))(1)
y = A× {1 + B × [exp (−K × t)]}**^−^**^1^(2)
y = A × {1 − B × [exp (−K × t)]}**^−^**^3^(3)
y = A × {1 − B × [exp (−K × t)]}(4)

The live weight (y, kg) equations at a given time (t) are considered the genetic and environmental factors. The parameter “A” represents the maximum possible weight of the animal regardless of these factors as the time approaches infinity. The “B” parameter indicates the inflection point of the growth curve and represents the rate of weight gain from birth to the maximum weight. The “K” parameter represents the relative growth rate and indicates whether the animal matures fast or slow. Higher values correspond to the animals that mature quickly, while lower values correspond to the animals that mature more slowly [7]. The degree of maturity (U) indicates the weight change in relation to the weight in adulthood, and it was predicted using the formula proposed by Lupi et al. [7] where the cumulative weight is denoted by y.
U = y/A(5)

Individual curves that failed to converge were regarded as outliers, and the parameter values for the model were ignored [15]. To compare the predictive performance of the tested models, several goodness of fit criteria were used, including the coefficient of determination (R^2^), the adjusted coefficient of determination (R^2^_adj_), the Akaike’s Information Criterion (AIC), the Schwarz Bayesian information criterion (BIC), and the residual mean square (RMS).
R^2^ = 1 − (RSS × TSS^−^^1^)(6)
R^2^_adj_ = 1 − (k − 1 × (k − n)**^−^**^1^) × (1 − R^2^) (7)
AIC = n ln (RSS × k^−^^1^) + 2k(8)
RMS = RSS × (k − n − 1)^−^^1^(9)
BIC = n × ln (RSS × k^−^^1^) + k × ln(n)(10)
where TSS and RSS are the total and residual sum of the squares, respectively, “k” is the number of parameters in the model, “n” is the number of observations, and “ln” is the log function [29,31]. The candidate model with the greatest R^2^ or R^2^_adj_ and the smallest RMS, AIC, and BIC values was selected as the best fit for defining the growth of animals. The AIC is a statistical measure for the comparative evaluation among time series models that provides an estimation of the information lost when a specific model is used to represent the process that generated the data. The BIC is a widely recognized method for model selection that prioritizes simpler models over more intricate ones. It considers the likelihood function and is similar to the AIC. The AIC and BIC were used to evaluate the adequacy of the models, where the number of estimated parameters is penalized. When presented with several comparable models, the most desirable model is the one with the lowest AIC and BIC scores [32], lowest RMS, and higher R^2^ or R^2^_adj_.

The models were fitted using the “easyreg R package” [30] in the R version 4.0.2 software (22 June 2020, The R Foundation, Vienna, Austria). The statistical significance was set at a *p*-value lower than 0.05.

## 3. Results

### 3.1. Exploratory Results

The PCA results showed that the first two principal components (PCs) accounted for 92.6% of the original variability. During the PCA, PC-1 did not show any clear segregation for w15–w150. However, w15, w30, w45, and w60 had positive PC-2 values, whereas w75, w90, w105, w120, w135, and w150 were grouped as negative PC-2 values, as shown in Figure 1. 

The scatter plotting for the samples grouped by year, sex, birth type, and season of birth showed no apparent outliers and an evident overlap of subgroups in each subgroup. This data is reported in the Appendix A. 

### 3.2. Nonlinear Growth Model Selection

All the proposed growth models (Gompertz, Logistic, von Bertalanffy, and Brody) showed similar goodness of fit criteria for predicting Kivircik lamb weight. Among all the models, the Brody model showed the lowest AIC, BIC, and RMS scores for all the subgroups, except for birth type. However, it is worth noting that for twin-born lambs, the Brody model failed to converge. The Logistic model was the less suitable model considering the highest AIC, BIC, and RMS scores for all the subgroups, except for single-born lambs (Table 3). 

The parameters (A, B, and K) estimated by the Gompertz, Logistic, von Bertalanffy, and Brody models for all the lamb, sex, birth type, and season of birth groups are reported in Table 4. The means ± SE for the A, B, and K parameters for all the lambs estimated by Gompertz were 38.52 kg ± 1.41 kg, 1.97 ± 0.02, and 0.01 + 0.001, respectively. The Logistic model estimated the A, B, and K values as 31.71 kg ± 0.67 kg, 4.36 ± 0.08, and 0.02 ± 0.001. The von Bertalanffy estimations of the A, B, and K values were 44.09 kg ± 2.21 kg, 0.51 ± 0.001, and 0.01 ± 0.0004; and the Brody model’s estimations were 105.27 kg ± 24.58 kg, 0.96 ± 0.01, and 0.002 ± 0.0004, respectively. The A values estimated by the Gompertz, Logistic, von Bertalanffy, and Brody models ranged between 35.24 kg (±1.51 kg) to 71.20 kg (±19.37 kg), 29.62 kg (±0.74 kg) to 35.04 kg (±1.27 kg), 39.37 kg (±1.53 kg) to 124.37 kg (±67.76 kg), and 75.35 kg (±15.34 kg) to 270.90 kg (±273.66 kg), respectively. Among all the models, it was observed that the Brody model overestimated the A values, which was biologically unrealistic. The same exaggeration of the A value was also observed using the von Bertalanffy model (A = 124.37 kg ± 67.76 kg for twin-born lambs). 

Combining the goodness of fitness criteria and biologically interpretable variables, our results revealed that the Gompertz model was the most suitable growth model for predicting Kivircik lamb growth. In the study, the Gompertz model predicted higher A and B values for male lambs (44.19 kg ± 2.88 kg and 2.07 ± 0.04), twin-born lambs (71.20 kg ± 19.37 kg and 2.61 ± 0.23), and winter-born lambs (38.48 kg ± 1.40 kg and 1.97 ± 0.02) compared to their counterparts. The K values were found to be similar for all the subgroups (0.01 ± 0.001). 

As shown in Figure 2, Figure 3, Figure 4 and Figure 5, the observed and predicted live weight values, estimation errors of the Gompertz model, as well as the degree of maturity for all the lamb, sex, birth type, and season of birth subgroups are presented, respectively.

According to the Gompertz model prediction, the lambs reached 40.7% of their mature weight at weaning (75 days of age) and at 150 days of age they reached 66.3% of their mature weight. Additionally, it should be noted that at 15 days of age, the Gompertz model showed a 0.30 kg error in predicting the live weight, which tended to improve up to the age of 60 days. However, between 75–120 days, the Gompertz prediction was observed to deteriorate up to 0.60 kg and at the age of 150 days, the Gompertz estimation was −0.43 kg (Figure 2a,b). 

During the fattening period, the female lambs showed a higher degree of maturity than the male lambs. The female lambs had a 5.8% greater maturity degree at weaning, and at 150 days of age the maturity gap was 6.4%. It was observed that the Gompertz model predicted the male lamb weights with a higher error compared to the female counterparts. At 120 days, the model had the highest error for both sexes by overestimating the male lambs’ weights at 1.47 kg and 0.53 kg for female lambs (Figure 3a,b).

Single-born lambs showed a higher maturity degree compared to twin-born lambs. At weaning, single-born lambs had a 41.2% maturity degree, while this was 31.9% for the twin-born lambs. At 150 days of age, single-born lambs were detected to have a 65.7% maturity degree and twin-born lambs a 60.7% maturity degree. However, it should be noted that the model showed an error of up to 18.77 kg for twin-born lambs, which was observed as the highest error among all the subgroups. Additionally, the model underestimated weight of single-born lambs from 15 to 150 days of age, ranging from 0.6 kg to 3.0 kg (Figure 4a,b).

In this study, spring-born lambs showed a higher degree of maturity than winter-born lambs. It was detected that the maturity gap between spring and winter-born lambs was closer among all the subgroups. At weaning, winter-born lambs showed a 39.4% maturity degree while this value was 41.0% for spring-born lambs. At 150 days of age, winter-born lambs were detected to have a 64.4% maturity degree and spring-born lambs a 65.6% maturity degree. The model underestimated the weights of spring-born lambs by up to 2.08 kg. Furthermore, the weights of winter-born lambs were underestimated by up to 1.63 kg (Figure 5a,b). 

## 4. Discussion

According to the results of this study, the mean birth weight and weight at slaughter (150 d) were determined as 4.36 kg (±0.86 kg) and 26.0 kg (±5.07 kg), respectively. Yakan et al. [33] reported similar values (4.34 kg) for the birth weight, but the lambs showed higher weight gains where Kivircik lambs reached approx. 26.0 kg at around 120 days. In a study by Alarslan and Aygün [34], the body weight at 150 days of age was determined as 33.68 kg (±0.5 kg). Similarly, Selvi and Üstüner [35] revealed higher lamb weights at 120 days of age (29.59 kg ± 6.78 kg) compared to this study. It is evident that in the current study, the lambs had lower slaughter weights compared to other studies by showing only a 66.3% maturity rate at 150 days of age. We believe the primary cause of the poor lamb growth could be attributed to the moderate growth performance of Kivircik lambs. Although it is not stated whether the lambs were pure or mixed with a Merino genotype in the mentioned studies, in our study, the lambs had a pure Kivircik genotype. Since Kivircik is an indigenous breed with a moderate growth performance, the growth of the lambs is expected to be poor without improving the genetic capacity. Therefore, aiding early intervention strategies, i.e., the selection of the best performing lamb(s) during pre-weaning, would improve the growth performance. Secondly, inappropriate management of the lambs particularly after weaning might cause the poor growth performance. After the weaning in our study, the lambs were sent to the pasture with their dams, which may have resulted in an inefficient use of the pasture that deteriorated the lamb growth. Yakan et al. [33] stated that after weaning, lambs were sent to the pasture without their dams while Alarslan and Aygün [34] mentioned that before weaning, the lambs were sent to the pasture at 2 months old with their dams to acclimate to the pasture. These management differences may also influence the growth performance of the lambs. 

Using mathematical functions to describe growth, most of the time growth is a homogeneous process [36]. However, Robertson [37] discovered cycles in the growth curves of body weight that distinguished growth into two or three phases. Zucker et al. [38] identified the causes of cyclic growth as resulting from changes in the environment and nutrition that accompany birth and weaning. Interestingly, the results of the PCA in the current study supported the multiphasic growth of Kivircik lambs, dividing the age−weight records into two dimensions, namely, pre-weaning and post-weaning dimensions. According to the results of the PCA, w15−w60 (pre-weaning phase) had positive PC2 values while w75−w150 (post-weaning phase) had negative values. The positive and negative loadings may be attributable to the fast and slow growth during the pre- and post-weaning phases. Boujenane [39] observed that, during the pre-weaning phase, the lambs presented faster growth, reaching up to 185 g/day, which was mainly related to the ewe’s milk yield. It is also worth noting that in the above-mentioned studies [33,34], the weaning age at 90 d and 150 d, respectively, was longer than the current study. Due to the fast growth during the pre-weaning phase, a longer weaning age may have also had a positive effect on the growth performance of Kivircik lambs. However, in opposition to the faster growth during pre-weaning, Bahreini Behzadi et al. [40], Ghavi Hossein-Zadeh [41], and Weber et al. [42] reported a declining growth rate starting from birth to the slaughter age. This was most probably due to inappropriate management techniques. 

In this study, the growth models for predicting the weight at a certain age in Kivircik lambs revealed similar goodness of fitness criteria, indicating that all the proposed models were statistically appropriate to use. However, some single-phase growth functions did not satisfactorily estimate the biological parameters even though they were statistically acceptable. We speculate that may been since the growth was divided into two dimensions, in shown in the case of Kivircik lambs. Overall, the Brody model had a similar adjusted R^2^ but the lowest AIC, BIC, and RMS scores for all the subgroups. However, when considering the biological parameters, namely the A values, the Brody and von Bertalanffy models overestimated this parameter. Furthermore, the Brody model did not converge for the birth-type subscale, which caused its application to be inaccurate. In particular, during the early phase of growth, the Brody model was reported to be inappropriate [7,10,11,43].

In this study, the Gompertz model with a sigmoidal portion of growth best described the growth of Kivircik lambs. Supporting our results, Yıldız et al. [25] revealed similar goodness of fitness criteria for predicting Kivircik lamb growth by indicating that the Gompertz function best described lamb growth. Furthermore, Paz et al. [44] indicated that the Gompertz model presented the best fit and biological interpretation in Morada Nova sheep for predicting the age−weight relationships. Waheed et al. [45] concluded that the Gompertz model with three parameters was appropriate for modeling growth curves in Thalli sheep. Keskin et al. [46] reported that the cubic model provided the best goodness of fitness criteria for predicting the growth performance of Konya Merino lambs. However, it overestimated the initial live weight. Therefore, they chose the Gompertz model for predicting the live weight at later ages from early partial life. Lewis et al. [10] stated that the Gompertz model provided a better fit for predicting the growth of Suffolk sheep. Topal et al. [47] determined that the Gompertz and Bertalanffy models showed the best fit for the growth of Morkaraman lambs. Malhado et al. [16] selected the Gompertz function to model the growth of local Brazilian sheep breeds. 

In the literature, it is evident that there is no consensus regarding one straightforward model over others that would predict lamb growth. Researchers have identified various models to describe the age−weight relationships in different breeds, but some models were also suggested for the same breed. Apart from the intrinsic factors, growth depends on environmental factors such as management, feeding, and health [13]. It is not advisable to expect data on actual growth, which may not reflect the actual potential due to deficiencies in feeding, climate, health, and stress, and may not be consistent with any form of growth function [10]. Therefore, these functions need to be tested prior to selecting the model. 

Growth functions are valuable tools that can reveal biologically important parameters, i.e., mature weight (A value), inflection age (B value), and growth rate (K value), which are reported to be heritable [6]. These parameters can be evaluated before applying a robust selection to improve the slaughter characteristics of animals since body weight and the rate of weight gain are the most economically important parameters for sheep fattening [12]. In our study, all the models predicted higher A values for male lambs compared to female lambs, which could be explained by sexual dimorphism resulting in heavier adult weights for male lambs [7,11,48]. In the current study, all the models predicted higher A values for twin-born lambs. Supporting our results, in the studies of Ali et al. [49] and Sharif et al. [13], the Brody model revealed the greatest A values for multiple-born lambs. However, in our study, the Gompertz model revealed a high estimation error of up to 16.8 kg, which should be considered when interpreting the high A values for twin-born lambs. In the current study, the lambs in the birth type subgroup showed an uneven distribution, which was acceptable since the Kivircik breed is not a prolific breed, and the high percentage of single-born lambs was compatible with the field conditions. However, this uneven distribution may have been the reason for the high estimation error of the Gompertz model. In terms of lambs born in winter, the Gompertz model predicted higher A values. The higher value of A in winter-born lambs may have been related to a lack of pasture use. While consuming grass in pastures, animals may experience energy deprivation due to increased physical activities and basal metabolism [50,51]. Similar to our results, Mokhtari et al. [52] reported higher A values for lambs born in winter.

The estimated B parameter using the Gompertz function was 1.97 for all the samples, where male lambs had higher values than female lambs, twin-born lambs had higher values than single-born lambs, and winter-born lambs had higher values than spring-born lambs. These findings indicate that male lambs, multiple-born lambs, and lambs born in winter may be lighter at birth, but the proportion of weight they may gain may be higher compared to their counterparts. Supporting our results, in the study of Boujenane [39], the best selected model (von Bertalanffy) revealed higher B values for twin or triplet Timahdite lambs.

Some studies revealed higher K values for females than males [7,11,48,53] while some determined similar K values [39,54]. In our study, almost all the models predicted similar K values for both sexes. Furthermore, we did not observe any differences in the K values between the birth type and season of birth in contrast to Boujenane [39] who found a higher maturation rate for single-born lambs.

While using the Gompertz model, it should be considered that the model tended to overestimate the live weights during the early ages and underestimate the final weights at slaughter. During the early stage of growth, the model had an error of up to 0.30 kg, the model’s prediction deteriorated between 75−120 days of age (up to 0.60 kg), and its prediction finalized at 0.43 kg less than the actual final weight. Gompertz, or any other function, cannot be expected to describe all the actual growth curves since animals may not show their potential growth due to environmental effects [9,10]. For instance, in this study, the lambs diverged from the actual slaughter weight (declared slaughter weight for the Kivircik lamb is 30−50 kg in the field by Turkpatent [20]) at 150 days of age by showing only a 66.3% maturity degree. As noted by Lewis et al. [10], the reported difference of 33.7% in the maturity rate at 150 days of age was most probably caused by inappropriate feed management/nutrition of the lambs. However, at this point, growth curves are shown to be good tools for following animal growth and identifying the causes of potential deviations from the expected growth performance. The utilization of a growth curve during the fattening of animals would allow for the implementation of early intervention strategies as soon as a deterioration in the maturity rate is observed.

## 5. Conclusions

The results of this study showed that the proposed growth models revealed closer goodness of fit criteria, indicating that the use of all four models (Gompertz, Logistic, von Bertalanffy and Brody) were appropriate for predicting Kivircik lamb weight. However, it should be noted that the Brody model did not converge and tended to overestimate the A values up to 270 kg. Considering the statistical and biological interpretations, the Gompertz function best described the Kivircik lambs’ growth. The mature weight and inflection age differed among the subgroups, but the growth rate was similar (K: 0.01 for all the subgroups). The male lambs (A: 44.19, B: 2.07), twin-born lambs (A: 71.20, B: 2.61), and lambs born in winter (A: 38.48, B: 1.97) were found to have a higher mature weight and the proportion of weight they may gain was higher compared to the female (A: 35.24, B: 1.91), single-born (A: 35.63, B: 1.88), and spring-born lambs (A: 35.89, B: 1.89). These differences in the A and B predictions should be considered when defining feeding management practices and the slaughter age for an economically profitable fattening performance. 

The Gompertz function provided ease in tracking the growth performance of Kivircik lambs. However, its estimation error should be considered when using the growth curve, especially for twin-born lambs. It was detected that the Gompertz model had an estimation error of up to 18.77 kg for twin-born lambs, which was the highest error among all the subgroups. Considering the results of this study, the Gompertz function proved to be a good tool for following Kivircik lamb growth, identifying any growth deterioration, and implementing early interventions possible as soon as deteriorations in the growth were detected. Additionally, the results may support selection programs for determining the best performing lambs during the early stages of growth. 

## Figures and Tables

**Figure 1 animals-13-02379-f001:**
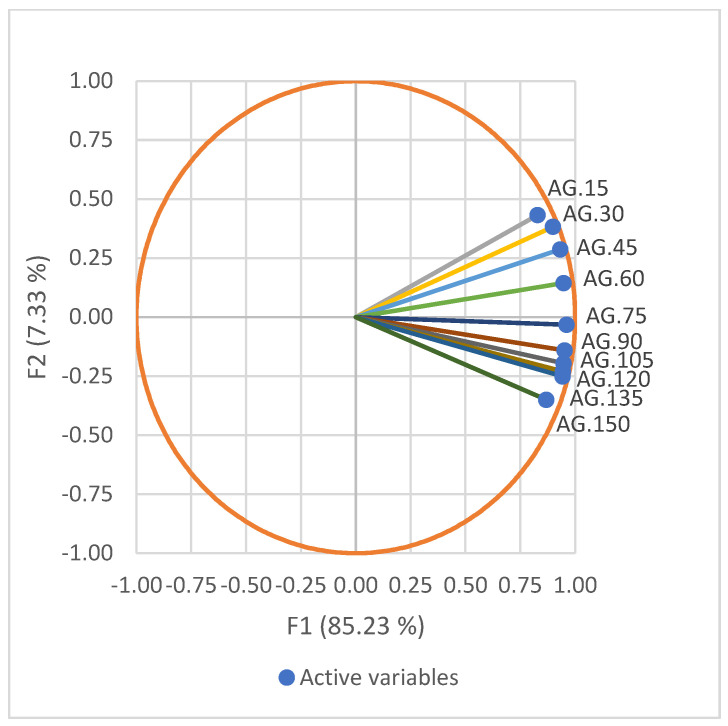
Projection of the ageweight records defined by the first two principal components (PCs). AGs (15–150) represents each weight variable between 15–150 days of age; F1 and F2 correspond to PC-1 and PC-2, respectively.

**Figure 2 animals-13-02379-f002:**
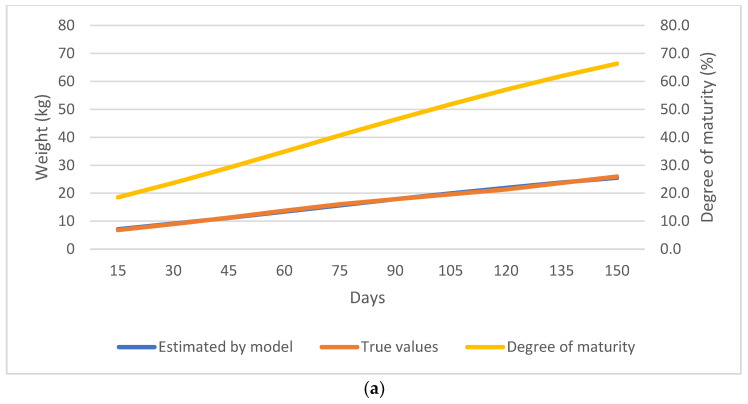
(**a**) Comparison of the true weight values and the values predicted by the Gompertz model. (**b**) Estimation error of the Gompertz model for all the samples.

**Figure 3 animals-13-02379-f003:**
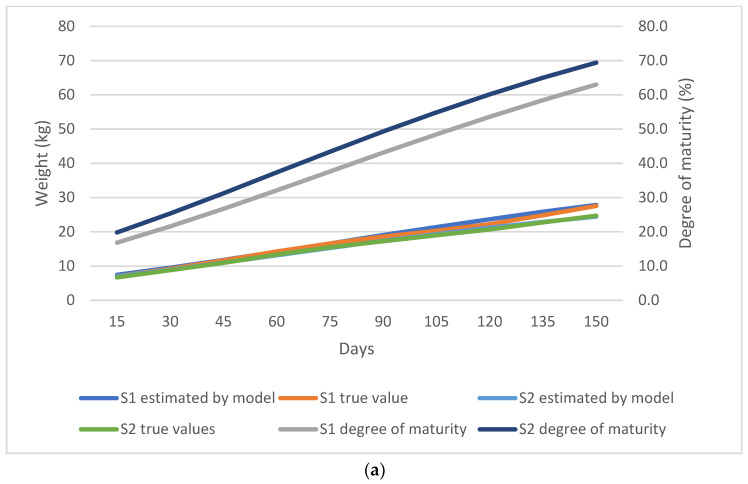
(**a**) Comparison of the true weight values and the values predicted by the Gompertz model for the sex subgroups; S1 denotes male while S2 denotes female lambs. (**b**) Estimation error of the Gompertz model for the sex subgroups; S1 denotes male while S2 denotes female lambs.

**Figure 4 animals-13-02379-f004:**
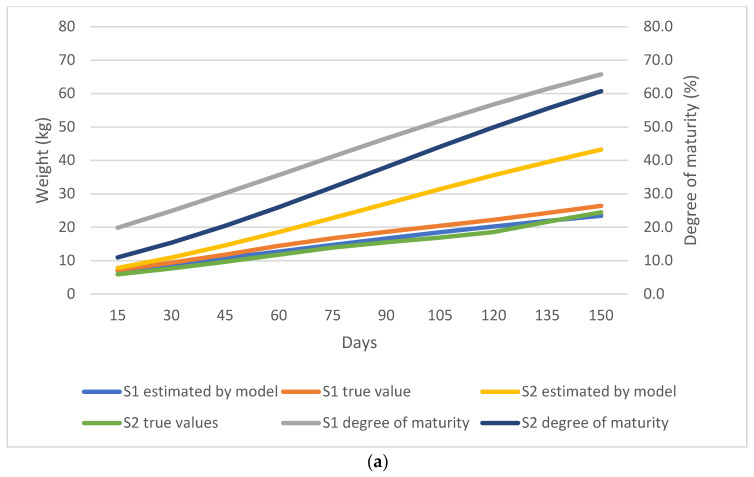
(**a**) Comparison of the true weight values and the values predicted by the Gompertz model for the birth-type subgroups; S1 denotes single-born lambs while S2 denotes twin-born lambs. (**b**) Estimation error of the Gompertz model for the birth-type subgroups; S1 denotes single-born lambs while S2 denotes twin-born lambs.

**Figure 5 animals-13-02379-f005:**
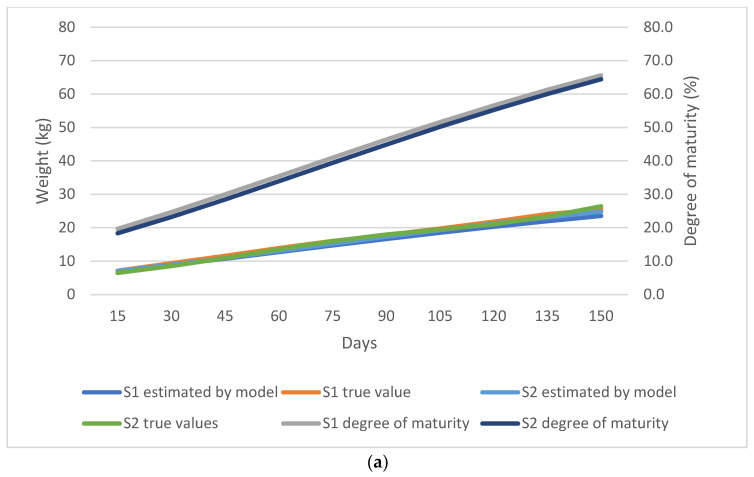
(**a**) Comparison of the true weight values and the values predicted by the Gompertz model for the season of birth subgroups; S1 denotes spring-born lambs while S2 denotes winter-born lambs. (**b**) Estimation error of the Gompertz model for the season of birth subgroups; S1 denotes spring-born lambs while S2 denotes winter-born lambs.

**Table 1 animals-13-02379-t001:** Descriptive statistics for the age–body weight records of Kivircik lambs.

Item	*n*	Mean ± Standard Deviation	Minimum	Maximum
Birth Weight, kg	536	4.36 ± 0.86	2.08	7.81
Weight at 15 d, kg	478	6.85 ± 1.34	3.31	11.34
Weight at 30 d, kg	608	8.98 ± 1.98	3.71	14.77
Weight at 45 d, kg	608	11.3 ± 2.50	4.50	18.36
Weight at 60 d, kg	607	13.8 ± 3.01	5.52	22.86
Weight at 75 d, kg	610	16.0 ± 3.43	5.97	26.37
Weight at 90 d, kg	600	17.9 ± 3.73	5.71	29.50
Weight at 105 d, kg	571	19.6 ± 4.04	5.34	32.90
Weight at 120 d, kg	550	21.4 ± 4.41	5.06	33.95
Weight at 135 d, kg	519	23.7 ± 4.61	10.00	38.81
Weight at 150 d, kg	498	26.0 ± 5.07	12.21	40.65

**Table 2 animals-13-02379-t002:** Frequency distribution of the lambs for the year, sex, birth type, and season of birth subgroups.

Variable	Subgroups	*n*	Relative Frequency (%)
Year	2014	261	42.6
	2015	104	17.0
	2016	247	40.4
Sex	Male	299	48.9
	Female	313	51.1
Birth type	Single	467	76.3
	Twin	145	23.7
Season of birth *	Winter	280	45.8
	Spring	332	54.2

* The lambs born in December, January, and February formed the winter subgroup while the lambs born in March, April, and May formed the spring subgroup. Both subgroups received the same amount of concentrate, but the winter-born lambs didn’t use the pasture.

**Table 3 animals-13-02379-t003:** Fit statistics for the nonlinear models for various subgroups of Kivircik lambs.

Variable	Fit Statistics	Nonlinear Models			
		Gompertz	Logistic	von Bertalanffy	Brody
All	R^2^_adj_	0.71	0.72	0.72	0.72
	AIC	30,960	30,994	30,950	30,921
	BIC	30,986	31,021	30,977	30,947
	RMS	13.5	13.6	13.5	13.4
	*p*-value	<0.001	<0.001	<0.001	<0.001
Sex					
	R^2^_adj_	0.73	0.72	0.73	0.73
Male	AIC	14,234	14,252	14,228	14,219
	BIC	14,257	14,276	14,252	14,243
	RMS	14.6	14.7	14.6	14.6
	*p*-value	<0.001	<0.001	<0.001	<0.001
Female	R^2^_adj_	0.73	0.72	0.73	0.73
	AIC	16,503	16,520	16,499	16,477
	BIC	16,527	16,544	16,523	16,502
	RMS	11.7	11.8	11.7	11.6
	*p*-value	<0.001	<0.001	<0.001	<0.001
Birth type					
	R^2^_adj_	0.75	0.75	0.75	0.73
	AIC	22,881	22,909	22,873	16,477
Single	BIC	22,906	22,935	22,899	16,501
	RMS	11.6	11.7	11.6	11.6
	*p*-value	<0.001	<0.001	<0.001	<0.001
	R^2^_adj_	0.68	0.67	0.68	-
Twin	AIC	7548	7554	7546	-
	BIC	7569	7575	7567	-
	RMS	14.7	14.8	14.7	-
	*p*-value	<0.001	<0.001	<0.001	-
Season of birth					
	R^2^_adj_	0.72	0.72	0.72	0.72
Spring	AIC	13,296	13,306	13,294	13,292
	BIC	13,320	13,329	13,318	13,315
	RMS	12.0	12.1	12.0	12.0
	*p*-value	<0.001	<0.001	<0.001	<0.001
Winter	R^2^_adj_	0.72	0.72	0.72	0.72
	AIC	30,945	30,980	30,936	30,921
	BIC	30,972	31,007	30,962	30,947
	RMS	13.5	13.5	13.4	13.4
	*p*-value	<0.001	<0.001	<0.001	<0.001

**Table 4 animals-13-02379-t004:** Estimated A, B, and K parameters for the various subgroups of Kivircik lambs.

Variable		Nonlinear Models			
		Gompertz	Logistic	von Bertalanffy	Brody
All	A	38.52 ± 1.41	31.71 ± 0.67	44.09 ± 2.21	105.27 ± 24.58
	B	1.97 ± 0.02	4.36 ± 0.08	0.51 ± 0.001	0.96 ± 0.01
	K	0.01 ± 0.001	0.02 ± 0.001	0.01 ± 0.0004	0.002 ± 0.0004
Sex					
	A	44.19 ± 2.88	35.04 ± 1.27	52.43 ± 4.86	270.90 ± 273.66
Male	B	2.07 ± 0.04	4.69 ± 4.69	0.53 ± 0.01	0.98 ± 0.02
	K	0.01 ± 0.001	0.02 ± 0.001	0.01 ± 0.001	0.001 ± 0.001
	A	35.24 ± 1.51	29.62 ± 0.74	39.59 ± 2.28	75.35 ± 15.34
Female	B	1.91 ± 0.03	4.15 ± 0.10	0.50 ± 0.01	0.94 ± 0.01
	K	0.01 ± 0.001	0.02 ± 0.001	0.01 ± 0.001	0.002 ± 0.001
Birth type					
	A	35.63 ± 1.05	30.59 ± 0.54	39.37 ± 1.53	75.35 ± 15.34
Single	B	1.88 ± 0.02	4.08 ± 0.08	0.49 ± 0.004	0.94 ± 0.01
	K	0.01 ± 0.001	0.02 ± 0.001	0.01 ± 0.0005	0.002 ± 0.001
	A	71.20 ± 19.37	41.96 ± 5.25	124.37 ± 67.76	-
Twin	B	2.61 ± 0.23	6.50 ± 0.72	0.66 ± 0.05	-
	K	0.01 ± 0.001	0.01 ± 0.001	0.003 ± 0.001	-
Season of birth					
	A	35.89 ± 1.73	30.09 ± 0.85	40.38 ± 2.63	78.53 ± 18.73
Spring	B	1.89 ± 0.03	4.05 ± 0.11	0.49 ± 0.01	0.94 ± 0.01
	K	0.01 ± 0.001	0.02 ± 0.001	0.01 ± 0.001	0.002 ± 0.001
	A	38.48 ± 1.40	31.70 ± 0.67	44.02 ± 2.20	105.27 ± 24.58
Winter	B	1.97 ± 0.02	4.35 ± 0.08	0.51 ± 0.01	0.96 ± 0.01
	K	0.01 ± 0.0005	0.02 ± 0.001	0.01 ± 0.001	0.002 ± 0.0004

## Data Availability

The data presented in this study are available upon reasonable request from the corresponding authors.

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
