# Peer review of "Comparison of Nonlinear Growth Models to Estimate Growth Curves in Kivircik Sheep under a Semi-Intensive Production System"

_animals, 2023, doi:10.3390/ani13142379_

Round 1

Reviewer 1 Report

I believe this paper has a very important audience and application and though it may seem fairly narrow, due to being an indigenous breed of sheep in Turkey, it may be a great springboard for others in various regions to follow as a roadmap. Overall, I believe that paper has a strong amount of merit and is very well written.

The study generally seems well thought out and designed, though I do have a couple of minor questions regarding some of the methodologies of live animal data and if that might impact your results.

- What was the age of the dams utilized in the study and was that accounted for at all in the analysis?

- Do you have any indication of dam milk production and do you believe this might alter the analysis?

- In the exploratory data, I noticed that there were a lot of lambs being reared as singles, do you have any explanation and do you believe that impacted your study at all?

- In your discussion, you mention references # 32 & # 33 and how your lambs were grown post-weaning as opposed to theirs. It might help you to briefly describe how the lambs in those 2 references were managed post-weaning.

I only found a couple of minor issues, primarily with verb tenses and usages.

Author Response

What was the age of the dams utilized in the study and was that accounted for at all in the analysis? Explained In the study, we did not include any affects related to the dams. Therefore, in our dataset, we don’t have any information regarding the dams’ ages.
Do you have any indication of dam milk production, and do you believe this might alter the analysis? Explained No, we didn’t measure the dam’s milk production. Therefore, we are not able to compare the growth performances considering the milk production of the dams.

In the exploratory data, I noticed that there were a lot of lambs being reared as singles, do you have any explanation, and do you believe that impacted your study at all?

Explained/done Kivircik sheep is not a prolific breed. Therefore, high percentage of single born lambs is expected. We think the unevenly distribution of the lambs in this subgroup may result in high error in Gompertz model estimation when estimating the birth type subgroup’s growth performance. This was included in line 397-401.
In your discussion, you mention references # 32 & # 33 and how your lambs were grown post-weaning as opposed to theirs. It might help you to briefly describe how the lambs in those 2 references were managed post-weaning. Done We thank reviewer for pointing out this fact. We have included the differences in postweaning management techniques among the studies in discussion. Furthermore, we have included the different weaning ages which affect the slaughter weights among studies. Line: 314-328 and line:344-348.
I only found a couple of minor issues, primarily with verb tenses and usages. Explained Considering the reviewer suggestion, our manuscript has been professionally proofread. We provide a certificate for your information.

Reviewer 2 Report

General remarks:

The topic of this manuscript is interesting for Animals’ readers, but I find some problematic parts in the text, mainly in the Materials and Methods section.

Details

Title: according to the low weight gain during fattening (10 kg during 75 days-133g/d) and fattening method (lambs were kept on pasture), I recommend a revision of the title: please add “Comparison… Kivircik Sheep Under Extensive Fattening or Under Extensive Production System!

Introduction:

Please add more info about the prevalence of the pasture based fattening method in Kivircik sheep!

Materials and Methods:

No information about mother ewes’ feeding system! See lines 279-293, need some data about the feeding of the mother ewes!

Please add more info about alfalfa hay and pasture, e.g. ratio of the crude protein!

All lambs after weaning were kept on pasture separate by their mothers? See lines 277-278 (“…lambs were sent to pasture with their dams…”!

The feeding was the same between two seasons? See lines 340-345, where the winter lambs were kept indoor (“indoor feeding”)! Please clarify it!

Author Response

Title: according to the low weight gain during fattening (10 kg during 75 days-133g/d) and fattening method (lambs were kept on pasture), I recommend a revision of the title: please add “Comparison… Kivircik Sheep Under Extensive Fattening or Under Extensive Production System! Done Title was changed as “Comparison… Kivircik Sheep Under semi-intensive Production System”. Because in our study, in the evening, lambs received concentrate feed and alfalfa hay after they come from pasture.
Please add more info about the prevalence of the pasture based fattening method in Kivircik sheep! Done Additional information related to prevalence of pasture use of the Kivircik lambs was provided in lines 81 to 82
No information about mother ewes’ feeding system! See lines 279-293, need some data about the feeding of the mother ewes! Done Additional information was provided in lines 117 to 118.
Please add more info about alfalfa hay and pasture, e.g. ratio of the crude protein! Done Additional information was provided in lines 118 to 120 and 125-131.
All lambs after weaning were kept on pasture separate by their mothers? See lines 277-278 (“…lambs were sent to pasture with their dams…”! Explained In the study, after the weaning, lambs were sent to pasture with their dams.
The feeding was the same between two seasons? See lines 340-345, where the winter lambs were kept indoor (“indoor feeding”)! Please clarify it! Done The concentrate supply was same in two seasons, but winter born lambs didn’t use the pasture. Additional information was provided, line148-149 as well as line 340-345 was revised, current line 405-408.

Reviewer 3 Report

The article " Comparison of nonlinear growth models to estimate growth curves in Kivircik Sheep" is nicely written article about different growth models in sheep. However, I have some small remarks that are mentioned in General and Specific comments.

General comment:

It would be good to explain in the introduction how you decided to use these four models.

Based on you selection criteria, Logistic model has the best R2, AIC and BIC values and you choose Gompertz model. Moreover, all models have similar selection criterion and what was the reason to compare them. Maine problem with your article is that the methodology of the selection of the models and what was presented in the results does not match. Moreover, the reason why you did not discuss at all Logistic model which is based on your criteria the best remains unclear.

Specific comments:

L 15: higher that what?

L17-19: Please define clear conclusion based on your results.

Table 1 is missing the number of the animals measured at each time point.

L 152: It would be good to know how many curves were regarded as outliers compared to the total amount of curves.

L196: Brody model failed to converge – what happened with the other models?

Figure 2, I think this kind of data presentation is very confusing. I would omit error data and have clear graph with two y -axis. Additionally, I do not understand why you represent percentages with two decimal places?

Figure 3, 4 and 5 – the same comment like for the Figure 2.

L276: „inappropriate management of the lambs particularly after the weaning“ – how these data used for further selection purposes?

Author Response

It would be good to explain in the introduction how you decided to use these four models.

Done An additional information was provided regarding the selection of these models in line 105- 109.

Based on you selection criteria, Logistic model has the best R2, AIC and BIC values and you choose Gompertz model. Moreover, all models have similar selection criterion and what was the reason to compare them. Maine problem with your article is that the methodology of the selection of the models and what was presented in the results does not match. Moreover, the reason why you did not discuss at all Logistic model which is based on your criteria the best remains unclear.

Explained We selected the best fitting model by considering highest R2, lowest, RMS, AIC, and BIC values (Line 196-198). Among all subgroups logistic model provided the similar or lower R2 values while having higher RMS, AIC and BIC values compared to other models. Therefore, this model was not selected as the best fitting model, and this was explained in line 221-222.

L 15: higher that what?

Done The sentences were rearranged. Line 16 and 17.

L17-19: Please define clear conclusion based on your results.

Done Considering the reviewer’s suggestion, conclusion was rearranged. Line 19-21.

Table 1 is missing the number of the animals measured at each time point.

Done Please see Table 1, page 3.

L 152: It would be good to know how many curves were regarded as outliers compared to the total amount of curves.

Explained There weren’t outliers. This is also supported by figures S1, S2, S3, and S4, where it can be seen that there is no evidence for visual sample outliers selection, Line 213-215.

L196: Brody model failed to converge – what happened with the other models?

Explained

All the other models converged correctly after iterations. However missing convergence was reported for other (Richrads) model:

(https://doi.org/10.1016/j.theriogenology.2012.11.031)

Figure 2, I think this kind of data presentation is very confusing. I would omit error data and have clear graph with two y -axis. Additionally, I do not understand why you represent percentages with two decimal places?

Done We have changed the figures according to the reviewer suggestion.

Figure 3, 4 and 5 – the same comment like for the Figure 2.

Done We have changed the figures according to the reviewer suggestion.

L276: „inappropriate management of the lambs particularly after the weaning“ – how these data used for further selection purposes?

Done and explained We think the primary cause for the poor lamb growth is low genetic capacity of Kivircik lambs. As stated in the text, in our study lambs had pure Kivircik genotype. We suggested for early intervention (i.e., selection of best performing lamb(s)) during preweaning to improve the growth of Kivircik lambs. This was explained in line 314-320.

Reviewer 4 Report

The authors compared non-linear growth models to evaluate the growth curves of Kivirchik sheep.
However, there are some remarks.
1. In the abstract, you need to write in more detail the number of animals examined. Specify the research results!
2. In the study materials, describe what the animals received as concentrated feed.
3. In the manuscript, I did not see the results of assessing the significance of the results obtained.
4. Figure 1. There is no interpretation of the graph data.
5. Figures with the results of the data, as additional, transferring the bulk of the manuscript.
6. Edit the conclusion.
All these comments are noted in the attached review file.

The level of English is sufficient. However, minor editorial corrections are required.

Author Response

In the abstract, you need to write in more detail the number of animals examined. Specify the research results! Done The total number of lambs used in the study was included in line 29. As well as the research results were specified line 34-36.
In the study materials, describe what the animals received as concentrated feed. Done The composition of the concentrate feed was provided line 121-122.
In the manuscript, I did not see the results of assessing the significance of the results obtained. Done Considering the reviewer’s suggestion, we provided the p values in Table 3., page 6-7.
Figure 1. There is no interpretation of the graph data. Explained Figure 1 was explained in line 204-208 and its interpretation can be found in line 337 and 343.
Figures with the results of the data, as additional, transferring the bulk of the manuscript. Explained Due to scatter plotting of the principal component analysis only shows that there are no evident outliers, they are less relevant with the results of the study, we prefer to remain them as in the supplementary file.
Edit the conclusion. Done Conclusion was rearranged considering the reviewer’s suggestions.
The level of English is sufficient. However, minor editorial corrections are required. Explained Considering the reviewer suggestion, our manuscript has been professionally proofread. We provide a certificate for your information.

Round 2

Reviewer 3 Report

Dear authors,

Could you please explain using four different models that predict growth curves similarly?

Additionally, based on your selection criteria,  you did not choose overall best model?

Moreover, presentation of the data is very poor in Figures with y-axis with decimal places, x-axis where the scaling is not consistent and presentation of the model and errors in different sizes and axis labels are sometimes presented with big and sometimes with small letters.

Author Response

We thank reviewer for his/her time to evaluate our paper. Our answers to the specific concerns are listed in the table and also can be found in the word file. 

Comments of reviewers on first version of manuscript Status Place of changes on new version of manuscript / Explanations

Reviewer3:

Could you please explain using four different models that predict growth curves similarly?

Explained The study's rationale considered the lack of knowledge on environmental factors' effects on the Kivircik lambs' growth patterns (L: 96-102). Therefore, the study aimed to evaluate the four nonlinear growth models (L: 102-106), whose utility in similar contexts was demonstrated in the literature. However, we didn't know which one best fit our data.

Additionally, based on your selection criteria,  you did not choose overall best model?

Explained We would like to indicate that in our study, the log regression resulted in less performance, as reported in L: 217-218 and discussed in L: 345-350. Even though it can be evidence that all models performed similarly, the biological interpretation of the A, B and K coefficients demonstrated that the Brody model overestimates the "A" parameter as well as von Bertalanffy partially does, as reported in L: 230-233, discussed in L: 350-353, and literature supported our findings. Considering our results, we would like to underline that Gompertz was the most suitable model (L: 234-236) which was discussed in L: 356-370, and our findings were supported by the cited literature. Finally, we concluded that Gompertz was the best-describing model for Kivircik lambs’ growth (L: 430-435).

Moreover, presentation of the data is very poor in Figures with y-axis with decimal places, x-axis where the scaling is not consistent and presentation of the model and errors in different sizes and axis labels are sometimes presented with big and sometimes with small letters.

Done We thank the reviewer for focusing our attention on the lowercase or uppercase letters and some mismatches in the graphs, which now have been fixed, please see page 9-12. We apologise for the error.

Round 3

Reviewer 3 Report

Dear authors,

In your article you proposed model selection: L185-186.

Based on this criteria best overall model is Brody, then von Bertalanffy, Gompertz and Logistic. If you choose third model as the best one and you claim that it is known for Brody model to overestimate A (L354-355) why did you took it at all? Additionally, it is not clear why you did not choose second or fourth model? Therefore, you suggested methodology based on model selection and you do not choose the best model based on your results.

Author Response

Dear reviewer, 

Besides the overestimated A values, the main reason for not selecting the Brody model as best describing lamb growth was because it did not converge for all the subgroups. On the other hand, the Von Bertalanffy model overestimated the A values for the birth-type subgroup (L351-354). Therefore, we selected the third best describing model. In the manuscript, we underline that we selected the Gompertz model due to its statistical and biological interpretation (L234-236) based on the results obtained.

As the present is a confirmative study of the models applied to a novel subject (lamb growth curve), initially, we didn't know either the lack of convergence of Brody nor the potential overestimation of Brody and Von Bertalanffy for the case of Kivircik lambs. We applied the four candidate models suggested by the literature and revealed those outcomes.

Thank you for your suggestions and comments to our study.